# The Mechanism of the Channel Opening in Channelrhodopsin-2: A Molecular Dynamics Simulation

**DOI:** 10.3390/ijms24065667

**Published:** 2023-03-16

**Authors:** Qi Xin, Wenying Zhang, Shuai Yuan

**Affiliations:** Chongqing Key Laboratory of Big Data for Bio Intelligence, Chongqing University of Posts and Telecommunications, Chongqing 400065, China

**Keywords:** Channelrhodopsin-2, molecular dynamics simulation, ion channel

## Abstract

Channelrhodopsin-2 (ChR2) has been one of the most important objects in the study of optogenetics. The retinal chromophore molecule absorbs photons and undergoes an isomerization reaction, which triggers the photocycle, resulting in a series of conformational changes. In this study, a series of intermediate structures (including D470, P500, P390-early, P390-late, and P520 states) of ChR2 in the photocycle were modeled, and molecular dynamics (MD) simulations were performed to elucidate the mechanism of ion channel opening of ChR2. The maximum absorption wavelength of these intermediates calculated by time-dependent density function theory (TD-DFT) is in general agreement with the experimental values, the distribution of water density gradually increases in the process of photocycle, and the radius of the ion channel is larger than 6 Å. All these results indicate that our structural models of the intermediates are reasonable. The evolution of protonation state of E90 during the photocycle is explained. E90 will deprotonate when the P390-early transforms into P390-late, in which the two conformations of P390-early and P390-late obtained from the simulations are consistent with the experimental descriptions. To validate the conductive P520 state, the potential mean force (PMF) of Na${}^{+}$ ions passing through the P520 intermediate was calculated by using steered molecular dynamics (SMD) simulation combined with umbrella sampling. The result shows that the Na${}^{+}$ ions passing through the channel with a very low energy barrier, especially in the central gate, is almost barrierless. This indicates that the channel is open in the P520 state.

## 1. Introduction

Optogenetics achieved control of neuronal activity by expressing modified highly photosensitive ion channels in the brain through protein engineering [1], followed by the target these ion channels with light. In 2005, Deisseroth and his colleagues [2] characterized the Channelrhodopsin-2 (ChR2) protein in the mouse brain, which activated neurons and modulated light-induced brain activity. Optogenetics is currently inextricably linked to the study of neuroscience, and it is urgent to investigate the structural mechanism of ChR2 as an important issue. Revealing the open-state structure of ChR2 and the process of the photocycle from dark to open is essential for the further development of its function.

There are three gating sites within ChR2 [3], which are extracellular gate (ECG), central gate (CG), and intracellular gate (ICG), respectively, controlling the opening of ion channels. In ChR2, the absorption of photons triggers the photoisomerization of retinal resulting in a series of conformational changes and proton transfer [4,5,6], which activates the photocycle, as shown in Figure 1. In dark-adapted ChR2 (D470, means the maximum absorption wavelength ($\lambda_{max}$) is 470 nm), the conformation of the chromophore retinal is in the form of all-*trans*, 15-*anti*, and its Schiff base is protonated. After illumination, the isomerization of the C13=C14 bond [7] will yield the first P500 ($\lambda_{max}$ is 500 nm) intermediate, at which point the chromophore is in the 13-*cis* and 15-*anti* conformation. Deprotonation of the Schiff base of P500 leads to the P390 intermediate, whose possible counter ions are either E123 or D253 [7]. In P390 state, there are two intermediates with marked differences [8,9], which are named P390-early and P390-late. Previously published studies on the relationship between two intermediates were still unclear. It was generally acknowledged [5,7,9,10,11] that P390-early allows only water to enter and P390-late already allows Na${}^{+}$ ions to traverse the channel. After the reprotonation of the Schiff base, the P390 intermediate evolves into the P520 intermediate [3,8], in which the D156 acts as a proton donor. Afterwards, P480 forms [12] with the retinal isomerizing to 13-*cis* and 15-*syn* conformation, and the protein walks into the desensitization phase. Eventually ChR2 returns to the D470 and the photocycle ends. As a highly conserved residue in channelrhodopsins, the E90 is difficult to find in the corresponding positions in the helix of other [13,14] homologous rhodopsin families. For example, in bacteriorhodopsin (BR) [13] D212, K216, and D85 can be matched against D253, K257, and E123 in ChR2, respectively. The two rhodopsins share similarities in properties and structure, but the corresponding E90 could not be found in BR. Meanwhile, it was observed that the mutants of E90R [15] alter the selectivity of ChR2. In a word, E90 is an exceedingly important residue that determines the opening of channel in ChR2. A much-debated question is whether or not E90 is protonated during the photocycle. On the one hand, it was observed that E90 is deprotonated during the photocycle due to the negative band at 1718 cm^−1^ in the carbonyl region, which belongs to deprotonated E90, was not observed in the E90Q mutant [16]. The deprotonation of E90 facilitates the hydration of helices, which can form hydrogen bonds with K93, disrupting the link between TM2 and TM7 and creating the conditions for a channel opening [16]. On the other hand, previous research established that the protonation of counter ions in CG in the *anti*-cycle is sufficient to make helices hydrated and E90 does not require deprotonation, whereas in their proposed *syn*-cycle, the marker bands for E90 deprotonation and C=N-*syn* occur instantaneously and are not time-resolved in C=N-*syn*, and thus E90 keeps protonated from the D470 to P520 in the anti-cycle [8]. Furthermore, two states of P390 called P390-early and P390-late were experimentally observed [9], and it is suggested that P390-late is already in the open-like state. As a result, it has always been interesting to know whether the proton transfer of E90 is associated with the formation of P390 intermediate.

In the absence of a crystal structure of the open state of ChR2, it is of great importance to study the opening process of ChR2 as well as the open state structure by modeling. Based on molecular dynamics simulations of the dark-adapted crystal structure of ChR2, we revealed the process of the photocycle from the closed state to the open state and obtained the simulated structures of D470-like, P500-like, P390-early-like, P390-late-like, and P520-like. However, we still referred to above-mentioned terms as D470, P500, P390-early, P390-late, and P520 when discussing these “intermediate-like” states throughout the paper in order to be consistent with experiment. The simulated structures will unveil the differences and connections of each intermediate structure and prove that the obtained P520 structure is open-like by means of steered molecular dynamics. In response to the controversy of the protonation state of E90 in the ChR2 photocycle, the analysis of the results obtained by establishing the corresponding membrane protein system simulations concluded that E90 deprotonates in the process from P390-early to P390-late, and that E90 deprotonation is a necessary condition for channel opening.

## 2. Results and Discussion

### 2.1. Structural Changes during the Opening of the Ion Channel

As shown in Figure 2a, it can be seen that the ion channel of ChR2 was located among TM1 (Transmembrane helix 1) , TM2, TM3, and TM7 [17,18]. The residues on the inner side of these helices were closely related to the formation of the channel. After analyzing the side chain positions of residues on the ECG and CG as well as the ICG, a reasonable approximation is that the gaps between these residue pairs may represent the ion channel. The channel between ECG and CG was mainly composed of TM2 and TM7, while ICG was primarily located between TM3 and TM7. The diameter of the channel directly reflects the degree of channel opening, and thus the diameter variation of channel with distances between residue pairs in the same plane was used to describe that indirectly. Experimentally, the minimum channel diameter in the open state is more than 6 Å [19]. As a result, a threshold 6 Å is labeled by black dashed line in Figure 2b, and the other calculated distances of residue pairs within the TM2, TM3, and TM7 s of all intermediates are displayed as well. It can be seen that the distances of the residue pairs near the CG of D470, P500, and P390-early states were less than 6 Å, implying that the CG of these states is blocked. However, all the distances of P390-late (orange line) and P520 (red line) intermediate states were larger than 6 Å, indicating that channel should be opened. Meanwhile, the distance of the residue pairs near the CG gradually increased from D470 to P520 intermediate, suggesting that the openness of the channel should increase as the photocycle proceeded. Taken together, these results prove that the position of the ion channel are portrayed very well. Certainly, potential conducting intermediates may be discovered preliminarily.

The E90, R120, and R268 [3] are the cores of ECG, CG, and ICG, respectively. Consequently, some key residues closed to them and most reported were chosen for structural descriptions. It can be found that the structural details of key residues in multiple intermediates during the photocycle are shown in Figure 3. In the D470 state, a complex hydrogen bonding network linked by K93 on TM2, D253 on TM7, E123 on TM3, and E90 on TM2 prevented water molecules from entering the CG. Therefore, the channel is incapable of being hydrated, which is not conducive to channel opening. Currently, the N atom on the Schiff base oriented towards the outer side of the membrane formed a hydrogen bond with E123. In the P500 intermediate, K93 still formed hydrogen bonds with D253 and E90. Like D470 state, a similar hydrogen bonding network forms between TM2, TM3, and TM7, which is not favorable to the formation of an opened channel as well. However, the N atom on the Schiff base was oriented toward the inner side of the membrane. This change is consistent with the results of previous studies [7]. In the P390-early intermediate, E90 formed a hydrogen bond with E123, K93 still formed a hydrogen bond with D253, and the hydrogen bonding network between TM2, TM3, and TM7 is still a disadvantage to form channel. In the P390-early stage, E123 received a proton from Schiff base, which broke the original hydrogen bond between E123 and LYR. The connection between TM3 and TM7 should be weakened due to the disappeared hydrogen bond in surface, and the channel has tendency to forming a conducting state; however, the T127 on TM3 formed a salt bridge with LYR. This strong electrostatic interaction, which counteracts the former weakening. As a result, the P390-early stage allows only the permeation of protons and a fraction of water molecules instead of forming an open state. In the P390-late intermediate, only K93, E90, and E123 formed a local hydrogen bonding network with each other, and there were no remarkable interactions between TM2 and TM3, TM2, and TM7. Accordingly, it is believed that the electronegativity of deprotonated E90 allows it to attract K93, E90, and E123 residues which originally block the entrance of the channel, transforming the globally complicated hydrogen bonding network into several locally small ones. In the P520 intermediate, the ECG and CG were further expanded to form a conductive channel favoring the passage of Na${}^{+}$ ions. The reason for the further opening of the ECG and CG at the P520 stage may be that D156 transfers protons to the Schiff base on the LYR and the reprotonation [20] is thought to be the key to the formation of photocurrents in channelrhodopins.

To qualitatively measure the degree of channel opening, we calculated the number of water molecules distributed in the CG of the channel (Figure 4). It is obvious that from P390-early to P520 intermediate, the number of water molecules at the CG shows a stepwise increase, corresponding to the “inactive, 70% is active, 100% is active” gating steps in ChR2 described by Lórenz-Fonfría [9]. Interestingly, the ratio of the number of water molecules in P390-late to that of P520 intermediate was also approximately 0.7, implying that the number of water molecules can be used to describe the degree of channel opening. With respect to the conformational changes of the ICG, the conformational changes of residues that were considered relatively important by previous reports [9] are displayed in Appendix A. The side chains of R268 and H134 block the ICG under the dark-adapted state. By comparison, the side chain of R268 is shifted substantially so that this residue no longer blocks the ICG in the P520 state. Compared with the D470 state, the position of the side chain of the H134 remains essentially unchanged, probably because the guanidine group on the arginine side chain is highly hydrophilic [21], making it easier to form a low energy path in the hydrophobic region at the inner gate. It also explains that the enhanced photocurrent and was observed in the H134R mutant of ChR2 [22], as the side chain of arginine is more prone to interact with the inner gate in the hydrophobic region compared to histidine, resulting in a greater degree of inner gate opening. The change of hydrogen bonding network at ECG allows the influx of water molecules and cations. The breaking of hydrogen bonds among LYR and TM2 and TM3 at the CG increases the aperture of the CG and allows the passage of Na${}^{+}$ ions and other cations, and the movement of R268 side chain at ICG further opens the originally blocked ICG. The structural changes of ECG, CG, and ICG indicate that the P520 obtained from equilibrium simulations is a relatively open structure. However, such a structure also has some problems. For example, the structure ignores the influence of other residues that form the channel during the channel opening.

### 2.2. Compared with C1C2 Crystal Structures

Recently, a series of crystal structures of C1C2 (delayed time points at 1 μs, 50 μs, 250 μs, 1 ms, and 4 ms) have been solved by using time-resolved serial femtosecond crystallography (TR-SFX) [23]. C1C2 is a chimera composed of ChR1(TM1-TM5) and ChR2(TM6-TM7). These resolved crystal structures of C1C2 provide a direct reference to validate intermediate conformations proposed for ChR2. According to the time scale of producing photocycle intermediates, the 250 μs, 1 ms, and 4 ms structures may correspond to our P390(-early and -late) and P520 models. However, conformational alignment showed that the structures of 250 μs, 1 ms, and 4 ms structures were almost superimposed identically in both spatial structures of proteins and retinal molecules, as shown in Appendix A. Therefore, the 4 ms structure was picked as a representative to compare with our intermediate models (P390-early, P390-late, and P520). The TM2, TM3, and TM7, which are vital to form the ion channel, were selected to compare, and the results of the partial conformation aligned are illustrated in Figure 5.

It could be noted that the three helices in P390-early intermediate are almost overlapping those in C1C2, suggesting that our simulated P390-early intermediate corresponds exactly to the 4 ms structure. As far as P390-late and P520 states, the TM2 and TM3 successively shift outwards about 1.5 Å and 2 Å, respectively. It indicates that it is a process of gradual opening of the ion channel from P390-early to P520 state, consistent with the water distributions in Figure 6. The P520 model should especially be a highly open structure because of the large shift of TM3 [23].

### 2.3. P390-Early and P390-Late

How the protonation state of E90 changes during channel opening has been controversial. Specific details have been discussed in introduction. Studies on other rhodopsin [24] pointed out that the conserved Glu/Asp at the CG deprotonates during proton transfer, and the dissociated proton moves along the direction of the electric field, converting the electrostatic potential energy into the mechanical energy needed for conformational change. This conclusion enlightened that the E90 deprotonation in ChR2 may play the same role. We simulated the proton transfer in E90 from P390-early to P390-late, and found that not only did the hydrogen bonding network at the ECG and CG of the channel change significantly, but also the number of water molecules at the CG in the P390-late channel was significantly higher than those in the P390-early intermediate. It is not unexpected to have such a result, which coincidentally indicates that E90 should deprotonate from P390-early to P390-late. The E90 deprotonation reasonably explains why the experimentally observed P390 also has an open-like structure. Furthermore, it is a mutual authentication of the experimental and simulated results when simulating the P390-early to P390-late. Note that the calculated $\lambda_{max}$ (see Section 3.5 for details) of P390-late is 302.67 nm, which is significantly blue-shifted compared to P390-early. Bond length comparison of the partial chromophore molecule of P390-early and P390-late is displayed in Figure 6. As can be seen from Figure 6, the most significant difference between the P390-early and P390-late structures is the length of the C14-C15 bond; the former is only 1.35 Å, which can be considered as a π bond, while this bond in the latter is obviously a σ bond. Therefore, blue-shifted absorption arose due to reduced π bonds in P390-late. In this subsection, it was proposed that the deprotonation of E90 occurs when the P390-early transfers to P390-late state, with a reasonable structural comparison corresponding to the experimental phenomena given. The deprotonation of E90 at this moment has a decisive role for the opening of the channel.

### 2.4. Verification of Channel Openness

ChR2 is a light-driven, non-selective cation channel in which the chromophore leads to the passage of scores of monovalent and divalent cations after absorbing photons [19]. The conformational details for the opening mechanisms of ion channels of ChR2 have been discussed in the above. Furthermore, the specific degrees of channel opening depend more on the free energy change of the ions passing through the channel. To verify that the channel of P520 intermediate is truly open, the water density analysis distribution and the potential mean force (PMF) calculations were performed. The distribution of water molecules within the channel is demonstrated in Appendix A. In the P520 intermediate, a great deal of water molecules fills the space at CG and ICG, forming a continuous water density distribution. Cations are mostly present in the form of hydrated ions [25] in water so that they are able to pass through the channel. The water density around the CG and ICG verifies that the opening of these two gates may occur simultaneously, because we did not change the state of the residues of the ICG when we simulated the photocycle. To understand the mechanism of the penetration of Na${}^{+}$ ions through the channel, the kinetics of the ionic transmembrane process in the P520 intermediate was simulated by SMD, and the PMF was calculated by umbrella sampling. The PMF of Na${}^{+}$ ions vividly reveals the ion free energy changes as a function of the coordinate of channel, which directly illustrates the open degree of the channel. In Figure 7, the absolute value of the, PMF was very low at all the three gating sites when the Na${}^{+}$ ions passed through the channel, reflecting the overall low resistance inside the P520 channel, and it can be basically assumed that the ion can pass quickly and freely at this intermediate. The free energy of the Na${}^{+}$ ions gradually decreased as the ions passed through the ECG, indicating that the channel is favorably conducive to the influx of Na${}^{+}$ ions. When the ion reached the CG, the free energy appears to decay to 0 kcal/mol, implying that Na${}^{+}$ ions passing through the CG is barrierless, which is evidence to support the fact that the P520 intermediate obtained from our MD simulations is exceptionally close to the open state. Compared to the CG, the free energy of Na${}^{+}$ ions at the ICG is slightly higher, which can be attributed to the fact that H134 may impede the efflux of Na${}^{+}$ ions. It is conspicuous that the free energy surface potential of Na${}^{+}$ ions through the channel generally conforms to an ideal curve of lower energy. The maximum energy barrier of P520 is reduced by an order of magnitude, compared to the value of the energy barrier derived from our previous work at the P500 stage [18,26]. In conclusion, the free energy potential of Na${}^{+}$ ions at the P520 intermediate reflected the energy change of Na${}^{+}$ ions through the channel, and the energy barrier at the CG of the channel decreased to 0 kcal/mol, which also bears testimony to the change of the hydrogen bonding network at the central gate as described above. Therefore, the simulated P520 intermediate model is an open channel or at least an open-like channel.

## 3. Materials and Methods

### 3.1. MD Simulations

Molecular dynamics (MD) as well as steered molecular dynamics (SMD) simulations were done with the [27,28] NAMD 2.13 software (University of Illinois at Urbana-Champaign, Champaign, IL, USA). The force field file describing the deprotonated LYR (Retinal+K257) referred to this document [29], which is available from this https://www.charmm.org/ubbthreads/ubbthreads.php?ubb=showflat&Number=38237#Post38237 (accessed on 26 March 2021), and the rest of the force field files were all based on CHARMM36 forcefield files [30]. This force field protocol was used by Kato et al. [31] in MD simulations for anion channelrhodopsins previously. The long-range electrostatic interactions were calculated using Particle Mesh Ewald (PME) method, and the truncation radius for the short-range non-bonding interactions was set to 12 Å. The SHAKE [32] algorithm was used to restrain the vibrations of the covalent bonds to all hydrogen atoms. The timestep was set to 2 fs. The steepest descent method was used to minimize the energy of the system under the NVT ensemble. The Langevin Piston algorithm [33] was used to keep the pressure at 1 atm, after which the system was equilibrated using NPT and harmonic restraints were applied to some dihedrals and planes of the protein, and the harmonic restraints were gradually reduced during equilibrium steps. Finally, unrestricted molecular dynamics simulations were performed under the NPT ensemble. From D470 to P520, three independent trajectories were run for each intermediate, and their root-mean-squared deviations (RMSDs) are shown in Appendix A. The equilibrated trajectories were first clustered to obtain the optimal conformations by k-means clustering, and the results of cluster analysis are listed in Appendix A and Appendix A. Then the cluster structures were aligned. We found that the structures of the same intermediate remained basically consistent, indicating that the correct setting of the protonation state of the key residues of the protein can produce convincing ChR2 intermediates. The SMD combined with the Umbrella Sampling method referred to this [34], and the whole process was kept as default except for setting the spring coefficient K = 4 kcal/mol × Å${}^{2}$ and velocity vs. = 0.00004 Å/step. The cluster analysis of the MD trajectories after RMSD equilibration and the water density maps were analyzed using AmberTools and Chimera [35,36]. The Python package matplotlib [37] package was used to create Figure 2, Figure 4, Figure 6 and Figure 7.

### 3.2. Modeling D470

The crystal structure of ChR2 (PDB ID: 6eid) [3] was obtained from the PDB database, the A chain was selected and embedded in a pre-equilibrated 16:0/18:1c9-palmitoyloleyl phosphatidylcholine (POPC) lipid bilayer to construct transmembrane structure of ChR2. The protonation states of several key amino acid residues were set as follows: Protonated Schiff base, deprotonated E123 and D253, and protonated E90 and D156. To simulate the protein in a physiological environment, we placed the protein and phospholipid bilayers in a periodic box (75 × 75 × 117 Å${}^{3}$) composed of TIP3 water molecules, which were adjusted to electroneutrality with 0.15 mol·L^−1^ NaCl solution. The generation of phospholipid bilayers and the assemble of the membrane protein system were achieved using the Membrane Builder on the CHARMM-GUI [38] online website. The whole system contains a total of 245 residues of amino acids 33-279 in ChR2, 13098 water molecules, and 138 phospholipid molecules, for a total of 61816 atoms. The equilibrated trajectory was analyzed by AmberTools, and cluster structures with statistically ranked percentages were generated. The conformation (cluster structure) with maximal percentage, was assigned to D470 state.

### 3.3. Modeling P500

Since MD simulations based on Newtonian mechanics cannot handle excited states, the isomerization of all-*trans* to 13-*cis* of retinal is difficult to simulate directly in MD strategy. In this study, the following procedures were used to obtain P500 conformation. First, starting from the D470 structure of ChR2, the C12-C13=C14-C15 dihedral was artificially tilted by 20° each step until this dihedral rotated to *cis* conformation. For each step, the dihedral was frozen and MD simulations was performed to remove the molecular tensions. Second, the consequential 13-*cis*,15-*anti* was simulated by MD unconstrainedly for 100 ns to obtain equilibrated conformation. Finally, the representative conformation of P500 was picked by the same method as that of D470. In our previous study [18,26], the 13-*cis* retinal was taken from the BR (PDB ID:1IXF) and was used to replace the all-*trans* retinal of ChR2 to obtain the P500 conformation after MD simulations. On account of the large conformational differences between all-*trans* retinal and its 13-*cis* isomer, direct replacement of retinal molecules might be caught up in a local potential well and far from the global minimum conformation. In this study, a stepwise tilting approach was used to gradually construct 13-*cis* conformation , with a smaller difference in conformational change at each step, which is more acceptable to obtain a stable conformation of P500. In fact, some previous investigations [7,8] have used the same method to model the *cis*-retinal of channelrhodopsins.

### 3.4. Modeling P390 and P520

According to the current experimental and theoretical literature [3,18], either E123 or D253, which is deprotonated, may act as counter ions to accept a proton from the Schiff base during the conversion of P500 into P390. In the P500 conformation, the electronegative E123 forms a salt bridge with the electropositive Schiff base with a distance of 1.6 Å, which is much smaller than that of 5.5 Å between D253 and the Schiff base (as seen in Appendix A). It is suggested that E123 is more possible to be a proton accepter and keeps protonated in P390 state, while the Schiff base is deprotonated. Therefore, the P390 model with protonated E123 and deprotonated Schiff base was constructed and performed 300 ns MD simulations to obtain thermodynamic equilibrium conformation. The representative conformation of P390 was selected by the same method as that of D470. This model differs from that of Kuhne’s research [8] in that they used the final structure of the P500 simulations with Schiff base deprotonated and D253 protonated. Moreover, we used a modified CHARMM36 force field [29] that is capable of describing the deprotonated retinal to deal with the membrane protein system instead of OPLS/AA force field used by Kuhne. Spectroscopic experiments [9] showed that P390 splits into P390-early and P390-late. As a result, to a substantial extent the above-mentioned P390 is P390-early. In fact, the channel is reportedly [9,11,39] to be pre-opened in P390-early and allowed protons and 30 percent of water to enter while the more open P390-late state allows sodium ions and 70 percent of water to immigrate. Obviously, the deprotonated E90 will provide more electronegative environment for sodium ions to pass through the channel. Since no details of the P390-late structure have been reported so far, we assumed that E90 loses its proton in the P390-late state [16]. In order to obtain P390-late state, the same measures that are used to simulate P390-early by MD were taken except that the E90 was set as deprotonated [16]. It was observed that the proton on D156 will be transferred to the Schiff base during the conversion of P390 into P520 intermediate [3,11,40]. Therefore, D156 transforms to deprotonated state, and the Schiff base reverts to protonated state in P520 state. After that, the membrane protein system was established based on these, and the MD simulations was run for 300 ns. The representative conformation of P520 was selected by the same method as that of D470. In summary, we set initial protonation states of relevant residues of each intermediate based on the explicitly known proton transfer reported in experiment. The set of protonation states of the key residues throughout the photocycle can be found in Appendix A. The pKa values of several key residues for each intermediate were calculated by Propka3 [41] and listed in Appendix A. The calculated pKa for E90, E123, D253, and D156 based on simulated conformations were qualitatively consistent with their protonation or deprotonation setting in D470, P500, and P390-early states, respectively. It suggests that our MD simulations obtain the right conformations, rather than obtaining other local minima.

### 3.5. Calculate Maximum Absorption Wavelength

To validate the conformation built in this research, the maximum absorption peak ($\lambda_{max}$) of LYR (Retinal + LYR) for each intermediate was calculated at TD-DFT/6-311++g(d, p) level in Gaussian 09 package [42]. The CAM-B3LYP function was verified to be suitable for calculating the vertical excitation energy of organic molecules [43] and was used in this research. In general, it is chromophore molecule that absorbs photons in protein; therefore, we only calculated the vertical excitation energy of retinal and covalently linked K257 approximately to reduce the computational cost, and the neglected impact of the protein environment did not introduce too many errors. The calculated $\lambda_{max}$ of each intermediate is listed in Appendix A. The $\lambda_{max}$ of D470, P500, P390-early, and P520 states are 463.67, 498.17, 369.48, and 510.58 nm, respectively, which are basically closed to the experimental values [3]. It is worth noting that the $\lambda_{max}$ of P390 state was experimentally resolved to be 375 nm [9].

## 4. Conclusions

In this study, we modeled the intermediate structures of D470, P500, P390-early, P390-late, and P520 of ChR2 during the photocycle by performing MD simulations. The spectral calculations at the TD-DFT level show that the maximum absorption wavelength of each intermediate is in general agreement with the experimental value, indicating that the conformations of the intermediates are reliable. It was found that ECG, CG, and ICG, which blocked the ion channel in the dark-adapted state, were eventually opened in the P520 intermediate, forming a conducting state. The comparison between C1C2 crystal structures solved by TR-SFX and our simulated intermediates provided a direct evidence for validating intermediate conformations proposed for ChR2. The simulations discovered that the E90 residue plays a crucial role in the photocycle. We proposed that E90 will deprotonate during P390-early transforming to P390-late intermediate, which is a vital process that promotes the opening of the channel. It was perceived that the hypothesis rationalizes the experimental observation of a certain degree of opening at P390-late. In a word, this investigation reveals the photocycle of ChR2 from the dark-adapted state to the conductive state. The PMF of Na${}^{+}$ ions in channel was calculated by SMD and umbrella sampling to check whether the simulated channel is permeable for ions. The result shows that Na${}^{+}$ ions passing through the CG of P520 intermediate is barrierless, suggesting that the P520 is most likely to be a fully open state. Here, we have obtained the P390-late and P520 intermediates conformations from MD simulations, and hope that a subsequent experiment will confirm this.

## Figures and Tables

**Figure 1 ijms-24-05667-f001:**
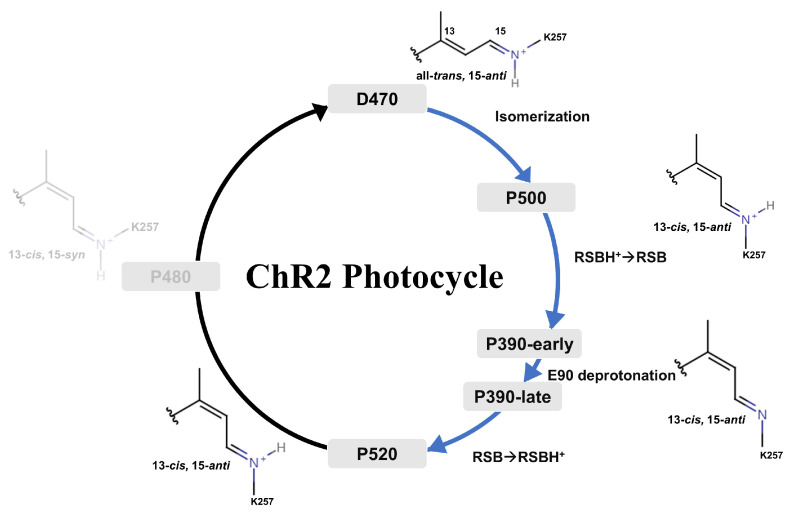
Schematic diagram of the photocycle of ChR2, and corresponding structural changes of retinal in the intermediate states during the photocycle. In the D470 state, retinal is in an all-*trans* conformation, and retinal is then isomerized to the 13-*cis* conformation by photoactivation to form the P500 state. The proton on the Schiff base is transferred to E123 or D253 to form the P390-early state. Subsequently, deprotonation of E90 leads to the formation of the P390-late state. Finally, the Schiff base is reprotonated to form the P520 state.

**Figure 2 ijms-24-05667-f002:**
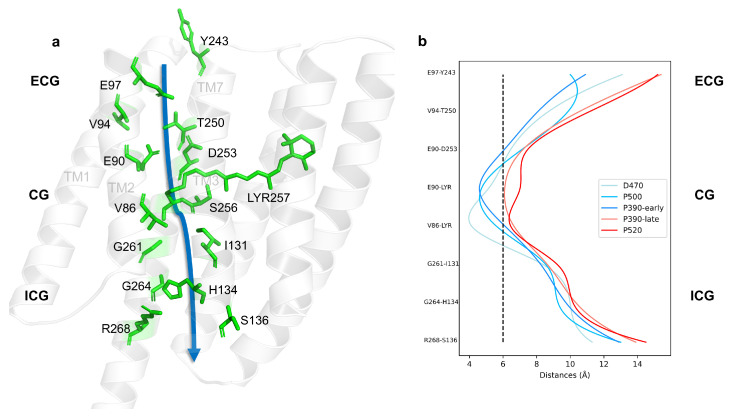
The outline of the ion channel in ChR2. (**a**) Key residues that forms the channel in ChR2. (**b**) The distances of inter-helix residue pairs along the channel. The vertical black dash line represents the experimental radius of an open channel.

**Figure 3 ijms-24-05667-f003:**
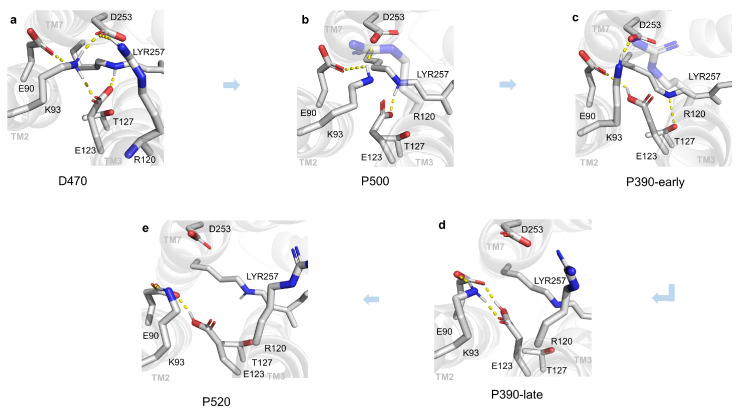
Structural changes of all intermediates in ECG and CG during photocycle. (**a**–**e**) exhibit the transformation of hydrogen bonding network in ECG and CG from D470 to P520 state and the originally blocked entrance of the ion channel turns into a nearly open pathway. LYR257 is the non-standard amino acid with K257 and retinal covalently-linked. The yellow dotted lines represent hydrogen bonds.

**Figure 4 ijms-24-05667-f004:**
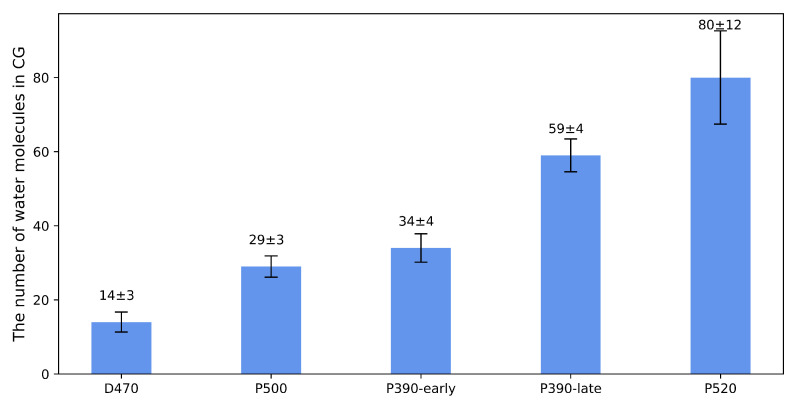
The number of water molecules at CG in all intermediates. The number of water molecules were counted by calculating the average number of all water molecules in CG in the last 50 ns trajectories in D470, P500, P390-early, P390-late, and P520 states.

**Figure 5 ijms-24-05667-f005:**
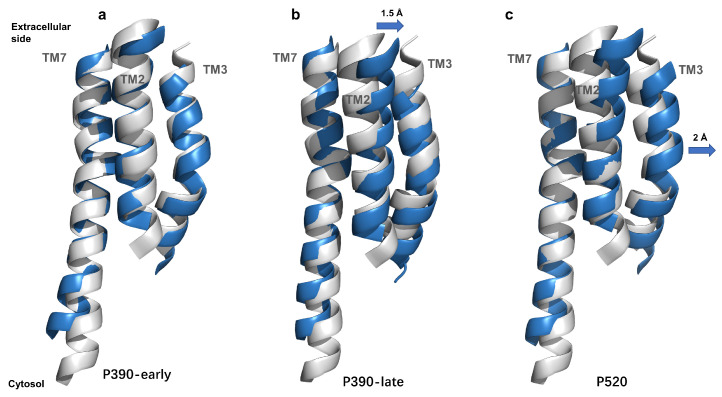
The superposition of C1C2 4ms crystal structure (gray) and simulated intermediates (blue). (**a**) P390-early state. (**b**) P390-late state. (**c**) P520 state. The blue arrows in (**b**,**c**) indicate the displacement of the helices of the simulated conformations over that of the C1C2 4 ms structure shifted outwards, respectively.

**Figure 6 ijms-24-05667-f006:**
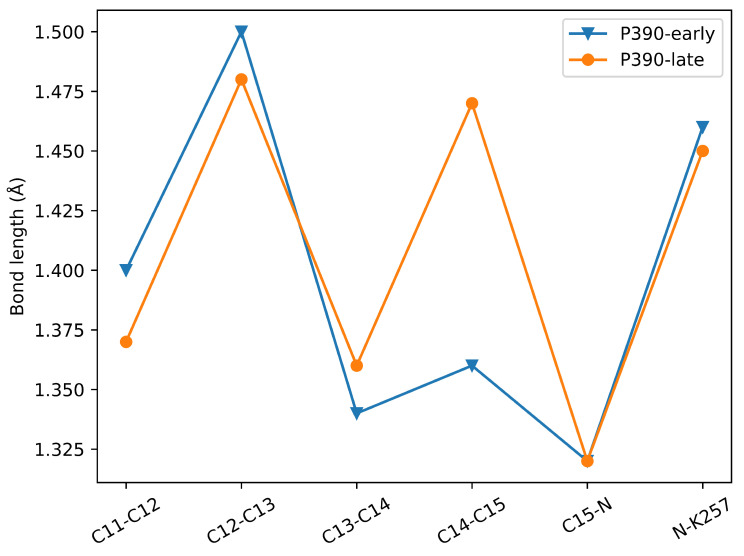
The partial bond length of retinal in P390-early and P390-late.

**Figure 7 ijms-24-05667-f007:**
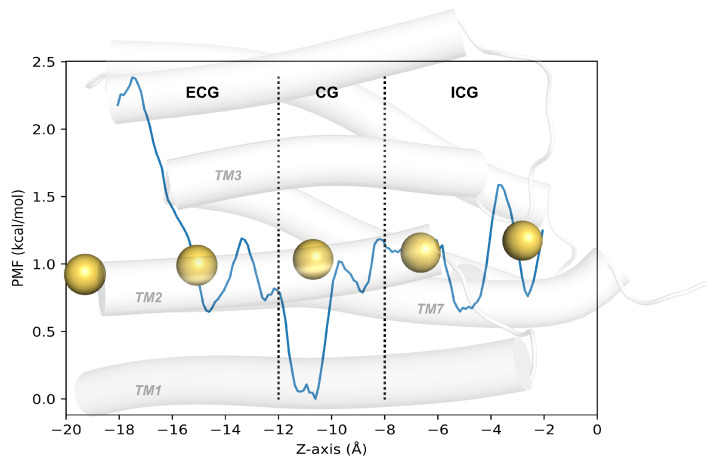
PMF of Na${}^{+}$ ions in P520 intermediate. The yellow sphere represents the sodium ion in motion.

## Data Availability

Not applicable.

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
