# Peer review of "The Mechanism of the Channel Opening in Channelrhodopsin-2: A Molecular Dynamics Simulation"

_ijms, 2023, doi:10.3390/ijms24065667_

Round 1

Reviewer 1 Report

This is an interesting qualitative work that computationally simulates the process of opening an ion channel in three gating sites within Channelrhodopsin-2 via the absorption of photons that triggers the photoisomerization resulting in a series of conformational changes and proton transfer. The authors computationally proved that at the end of the cycle, in the stage of P520 intermediate, all three gates are opened sufficiently to form a conductive channel for passing the Na+ ions. This work is very detailed; every stage of photocycle is described meticulously.  The simulations have been carried out exploiting state-of-the-art molecular dynamics modeling packages. In particular, to validate the conductivity of the ending state, the authors resorted to the steered molecular dynamics simulation combined with umbrella sampling. The manuscript is clearly written and well structured. In general, this work makes a good impression and I have no objections against accepting this paper in current form.

Author Response

We sincerely thank the reviewer.

Reviewer 2 Report

Manuscript ID: ijms-2235680

The manuscript reports a study of the opening mechanisms of rhodopsin channels. This in silico study is based on 3D modelling, MD simulations, DFT and an umbrella sampling of protein models representing rhodopsin intermediates that can correspond to excitation states at different wavelengths. The idea and the strategy chosen for such a study are good, however, there are a number of annoying drawbacks.

General Remarks:

The most important problem is the limited and superficial analysis of the generated date.

1. The conclusion on the similarity of conformations generated for a given protein state was only made using RMSD analysis. Cluster analysis will be much more efficient in obtaining the most populated conformation for each state. Moreover, the RMSDs calculated on the limited MD simulation (300 ns) showed that all systems are not really equilibrated. For such an ambitious project, the extended MD simulation is highly recommended.

2. What are the protein conformations shown in the figure, mean, final or randomly chosen frame?

3. H-bonds are not correctly characterized. Their populations (occurrences) over the simulation time should be reported along with their values and the calculated errors.

4. Similar for the water content in all intermediates studied.

5. Covalent bonds should be reported either as the mean values with rms errors or measured on the most populated conformations.

Additionally, the results can be nicely represented as a free energy landscape built on the reaction coordinates (the optimal specific metrics for this protein) for the concatenated data to visualize the difference between the states.

Particular remarks:

Abstract

Line 1: "tool" may be better "object" or "entity";

Line 4: D470, P500, …, which are the intermediate states corresponding to the excitation by different wavelengths must be explained and probably noted differently (for example, by introducing symbol/index S, for state) to avoid confusion with the identical representation of the protein residues across text (e.g. E123, D253, ….)

Line 5: “opening” should be referred to as “ion channel opening”

Line 12: All kinds of molecular dynamics simulations used in the study should be mentioned.

Introduction

The legend for Fig. 1 needs to be completed for the clarity.

Line 42: and everywhere in the text: Na+ must be completed by 'ion'

Line 84: “TM11, TM2…” must be preceded by “transmembrane helices”.

Line 59: 'for channel formation' should be better 'a channel opening'.

Fig. 2. The general title is missing. Also, this Figure must be placed after its first citation in the text.

Fig. 3 shows not only the structural difference between the intermediates, but also the non-covalent interactions. This should be reflected in the legend. The figure should be placed after its first citation. A LYR267 term is never decoded in the text or in the legend.

Fig. 4. How were the water molecules calculated? What conformations were used? A single conformation or a statistically valid conformational ensemble?

Line 139: “70 and 100 percentages” should change to 70% to 100%

Lines 154-156: The calculation of the cross-correlation matrix will be very helpful for such interpretation/conclusions.

Line 168: What is C1C2??

Fig. 5 should be placed after its first citation in the text.  Also, the caption needs to be improved. In fact, it shows the superposition of structures but not the structural comparison. Which conformations were chosen for overlay, middle, last or randomly picked?

Fig. 6. What is the error value in the bond length calculation?

Fig. 7. Meaning the negative "position". Negative about what?

All text should be carefully edited for a logic, accuracy of presentation and English spelling.

Round 2

Reviewer 2 Report

The  manuscript has been considerably improved by the Authors. 

Note: Each figure should be placed after its first citation in the text.